# Carotenoid Biosynthesis and Plastid Development in Plants: The Role of Light

**DOI:** 10.3390/ijms22031184

**Published:** 2021-01-26

**Authors:** Rocio Quian-Ulloa, Claudia Stange

**Affiliations:** Departamento de Biología, Facultad de Ciencias, Universidad de Chile, Las Palmeras 3425, Ñuñoa, Santiago 7800003, Chile; rocio.quian.ulloa@gmail.com

**Keywords:** carotenoids, photoreceptors, chloroplasts, chromoplasts, light signaling

## Abstract

Light is an important cue that stimulates both plastid development and biosynthesis of carotenoids in plants. During photomorphogenesis or de-etiolation, photoreceptors are activated and molecular factors for carotenoid and chlorophyll biosynthesis are induced thereof. In fruits, light is absorbed by *chloroplasts* in the early stages of ripening, which allows a gradual synthesis of carotenoids in the peel and pulp with the onset of chromoplasts’ development. In roots, only a fraction of light reaches this tissue, which is not required for carotenoid synthesis, but it is essential for root development. When exposed to light, roots start greening due to chloroplast development. However, the colored taproot of carrot grown underground presents a high carotenoid accumulation together with chromoplast development, similar to citrus fruits during ripening. Interestingly, total carotenoid levels decrease in carrots roots when illuminated and develop chloroplasts, similar to normal roots exposed to light. The recent findings of the effect of light quality upon the induction of molecular factors involved in carotenoid synthesis in leaves, fruit, and roots are discussed, aiming to propose consensus mechanisms in order to contribute to the understanding of carotenoid synthesis regulation by light in plants.

## 1. Introduction

Plants, as sessile and photosynthetic organisms, have to constantly adjust their growth and development in response to the environment. Sunlight is the main source of energy that regulates plant development after germination. Once the seeds germinate and the new seedlings sense the light, the photomorphogenesis program begins. Photomorphogenesis includes the de-etiolation process in which plants inhibit excessive hypocotyl growth and expand their cotyledons together with the differentiation of etioplasts into chloroplasts with concomitant high production of chlorophylls and carotenoids [1].

Carotenoids are plastidial isoprenoid pigments essential for plant life. They represent a numerous family of compounds in which over 850 natural carotenoids have been reported to date. Carotenoids are synthesized by plants, algae, some bacteria, and fungi [2,3,4]. In plants, they provide yellow, orange, and red colors to flowers [5], fruits [6], and some roots. Moreover, they participate actively in pollination and seed dispersion [7]. In chloroplasts enriched tissues, such as leaves, carotenoids fulfill different functions; they act as accessory light-harvesting pigments expanding the range of light absorption during photosynthesis and also have a photoprotective role protecting the light-harvesting complex, membranes, and proteins from excessive light incidence through thermal dissipation and free radical detoxification [8,9,10,11]. They also supply substrates for the biosynthesis of key plant growth regulators such as abscisic acid (ABA) and strigolactones [12,13,14]. Carotenoids also play relevant roles in human nutrition and health as antioxidants and an essential source of retinal (the main visual pigment), retinol (vitamin A), and retinoic acid synthesis, which are required for several biological processes including vision, embryo development, activation of the immune system, among others [15,16,17,18,19,20,21].

Despite the relevance of carotenoids for agriculture and human health, we still require further evidence to elucidate how carotenoid synthesis and accumulation are regulated in plant organs. So far, it has been established that carotenoid accumulation in certain organs depends on the rates of synthesis and degradation that are regulated at transcriptional and post-transcriptional levels in different plant organs, i.e., in photosynthetic tissues, fruits, flowers, seeds, and roots [22]. All mechanisms are tightly coordinated by both internal (senescence, circadian clock, epigenetic mechanisms, and ABA feedback) and external signals such as light [22,23,24]. 

In this article, we focus on the molecular mechanisms by which light signals are sensed by photoreceptors in different plant organs. We address the strategy to transduce the light signal at the molecular level in order to regulate the carotenoid metabolism—biosynthesis and accumulation. Moreover, we will also discuss the role of factors regulated both positively and negatively by light aiming to compare the signaling mechanism that affects carotenoid synthesis and storage in chloroplasts and chromoplasts of different tissues. In particular, we cover three case studies involving different scenarios of light-regulated carotenoid accumulation—leaves, fruits, and roots.

## 2. Phytochromes: The First Stage of Light Perception

Plant photoreceptors include the family of phytochromes PHYA-PHYE (PHYA, PHYB, PHYC, PHYD and PHYEs) that absorb light in the red (R) and far red (FR) range, and cryptochromes (CRYs) and phototropins that absorb in the blue and UV-A range [25,26,27]. Phytochromes (PHYs) are holoprotein of 125-kDA with a covalently attached tetrapyrrole (bilin) chromophore that senses the light and triggers the conformational switch to activate the protein [28]. The chromophore of the protein is located at the N-terminal while the light signal transduction domain is at the C-terminal of the PHYs [29,30,31,32,33].

PHYA belongs to type I photoreceptors being more labile to white (W) light, while PHYB-PHYE are more stable to W light [34]. Since PHYA and PHYB have been described as the phytochromes predominantly involved in the regulation of photomorphogenesis and carotenoid synthesis [35,36,37,38], we refer to PHYA and PHYB throughout the review.

PHYs can be found in two interconvertible forms—an active (Pfr) and an inactive (Pr) form. The Pr isoform absorbs light at 660 nm (R light), resulting in its transformation to the Pfr isoform that absorbs light radiation at 730 nm (FR light). Once Pr is activated (Pr isoform), it is translocated to the nucleus as a Pfr homodimer or heterodimer [27,39,40,41,42], accumulating in nuclear speckles [43,44]. PHYA is active and stable in FR light and in some R light intensities, participating in the high-irradiance response (HIR) and very low-fluence response (VLFR, 1 to 10 µmol m-2 of R light) where PHYA can favor seed germination [32,45]. Usually, PHYA is labile in R light, and the first 6 to 12 aa (enriched in serine) at the N-terminal part of the protein could be involved in controlling the degradation of PHYA [46]. On the contrary, PHYB is active only in R light participating in low-fluence response (LFR) [34,47]. Active PHYA is transported into the nucleus through far-red elongated hypocotyl 1 (FHY1) and its homolog FHY1-like (FHL) chaperons [48]. These proteins contain a nuclear localization signal (NLS), nuclear export signal (NES), and PHYA binding site [35,49,50,51]. For PHYB, two mechanisms of importation into the nucleus have been proposed; (1) through a masked nuclear localization signal (NLS) located in the C-terminal [31,52] and (2) through the help of phytochrome interacting factors (PIFs) transcription factor acting as chaperons. This evidence was suggested by studies in the unicellular green algae *Acetabularia acetabulum* L., where different PIF, such as PIF13, PIF4 and PIF5, which binds to PHYB through an active PHYB-binding (APB) motif and then translocates the photoreceptor into the nucleus [53].

Once PHYs are in the nucleus, they promote the phosphorylation of PIFs factors and stabilize transcription factors such as long hypocotyl 5 (HY5), long hypocotyl in far red1 (HFR1), and long after far-red light 1 (LAF1). These factors then bind to light-responsive elements (LREs) located in the promoter of genes involved in de-etiolation, photomorphogenesis [29,54,55,56,57], plant metabolism, signaling, stress defense, photosynthesis, and chloroplast development together with chlorophyll and carotenoids synthesis [58,59,60].

## 3. Carotenoid Synthesis in Plants

In plants, carotenoid enzymes are encoded by nuclear genes. Initially, the protein products are targeted as pre-proteins to the plastids, where they are post-translationally processed to become a fully functional enzyme. Carotenoid synthesis starts with the condensation of isopentenyl diphosphate (IPP) with the isomer dimethylallyl pyrophosphate (DMAPP), by means of geranylgeranyl pyrophosphate (GGPP) synthase (GGPPS) to produce GGPP. The formation of the first carotenoid, named phytoene, is produced from two molecules of GGPP and catalyzed by phytoene synthase (PSY), the most regulated step of the pathway [61,62,63]. The biosynthesis of carotenoids continues with several desaturations (z-carotene desaturase (ZDS), phytoene desaturase (PDS), isomerizations (z-isomerase (Z-ISO) and carotenoid isomerase (CRTiso)), and cyclations (lycopene β-cyclase (LCYB) and lycopene ε-cyclase (LCYE)) to produce several colored carotenoids including lycopene and a- and b-carotene (Figure 1) [64,65].

As mentioned before, carotenoid synthesis is regulated by different signals, either internal or exogenous, allowing their accumulation in chloroplasts and chromoplasts of several organs such as leaves, fruits, and some roots, etc. [22,23]. 

### 3.1. Leaves: Carotenoids Synthesis Takes Place in Chloroplasts

In photosynthetically active organs, such as leaves, chloroplasts accumulate chlorophyll a, chlorophyll b, and carotenoids (e.g., xanthophylls and β-carotene). These compounds are specifically located in light-harvesting complexes, fulfilling important functions during photosynthesis [66,67,68,69,70].

Throughout de-etiolation (plants that were transferred to light after being in darkness), carotenoid and chlorophyll biosynthesis and chloroplasts development are regulated in a coordinated manner [71]. When the plant is kept in dark conditions, photoreceptors are inactive in the cytoplasm [72,73], PIFs are stabilized by the DET1/DDB1/CUK4 complex allowing that PIFs bind to the LRE of PSY promoter inhibiting its expression and inhibiting carotenoid synthesis thereof (Figure 2) [74,75,76,77]. In dark-grown seedlings, PIFs also repress the expression of genes required for chlorophyll biosynthesis (e.g., *AtPOR*) and chloroplast development as well as promote the hypocotyl-elongation that helps the plant in searching for light [78,79]. In addition, constitutively photomorphogenic1 (COP1)/DDB1/CUL4 complex binds directly and triggers degradation of the basic helix-loop-helix (bHLH) HY5 and HFR1 transcription factors, which have a positive role in light signaling. Therefore, HY5 has an antagonistic role to PIF1 during photomorphogenesis (Figure 2) [80,81,82,83,84].

On the other hand, during de-etiolation or when plants are grown in W light (high R/FR ratio), PHYB is active [73,85] and physically interacts with COP1 in the nucleus, decreasing the protein ubiquitin ligase E3 activity [86,87] and therefore allowing HY5 to be released from the recruitment of COP1/DDB1/CUL4 complex (Figure 2) [80]. In addition, PHYB phosphorylates PIFs, leading to its degradation by the 26S proteasome [29,88]. These two main activities of PHYB promote the accumulation of HY5, which then can bind to the LRE (e.g., E and G-boxes) in the PSY promoter. Consequently, the carotenoid biosynthesis is induced, leading to an optimal transition to photosynthetic metabolism (Figure 2) [75,89]. Although the *BCHX2* carotenogenic gene has also a G-box in its promoter, it has not been described as a target of PIF1 [89]. Consistently, the sole upregulation of *PSY* expression is sufficient to increase the carotenoid synthesis in several plants, including de-etiolated seedlings [90,91,92]. However, carotenoid biosynthesis in the dark can also be induced by blocking gibberellic acid (GA) biosynthesis [89,91] because GA negatively regulates DELLA proteins, which are negative regulators of PIFs.

On the other hand, when plants are grown in the shade (low R/FR ratio), for instance, when they grow under a dense canopy, they perceive higher levels of FR due to the selective light reflection on the foliage. This causes a decrease in the carotenoid synthesis compared to W light [93,94] and *PSY* expression, including different FR intensities such as moderate shade (R/FR: 0.7) or severe shade (R/FR: 0.2) [94]. At the molecular level, active PHYA physically interacts with PIFs and COP1 in the nucleus, fulfilling the same function that PHYB carries out in W light (Figure 2) [86,95]. Additionally, PHYA interacts with HY5 and far-red elongated hypocotyls 3 (FHY3) through direct binding to modulate gene transcription. Interestingly, by means of ChIP-seq and RNA-seq of plants exposed to FR light, genes associated and regulated by PHYA were identified, including the key carotenogenic gene *PSY* [54,95]. This suggests a direct role of PHYA to promote *PSY* expression. Moreover, in shade, FHY3 and far-red-impaired response 1 (FAR1) promote the expression of phytochrome rapidly regulated 1 (PAR1) and its paralog PAR2 [96], which encode for transcriptional co-factors that bind to PIFs, sequestering them and allowing the *PSY* expression (Figure 2) [97]. Nevertheless, some PIFs remain stable, such as PIF4 and PIF5, which can stay bound to the LRE motifs, blocking them for HY5 binding [98,99]. Overall, the regulation of carotenoid synthesis in shade (including moderate and severe shade) is very complex and involves a delicate balance between the expression and repression of photomorphogenic genes including *PSY*.

### 3.2. Fruits: Carotenoid Synthesis Takes Place in Chromoplasts

Fruits are sink organs that store mainly sugars and pigments such as carotenoids [100], which accumulate at high levels in chromoplasts in crystals-like structures or plastoglobules [101,102,103,104].

Particularly, tomato (*Solanum lycopersicum* L.) is widely used as a fruit model for carotenoid synthesis [105]. Lycopene is the most abundant carotenoid in this fruit, and their synthesis correlates with *SlPSY1* expression [106,107,108]. Tomato has three *SlPSY* paralogs—*SlPSY1* predominantly expressed in fruits and petals, *SlPSY2* in leaves, sepals, and petals, and *SlPSY3* in roots, particularly after deprivation of phosphate and mycorrhization [108,109,110,111]. 

In tomato fruits, phytochrome regulates *SlPSY1* expression in fruits and the activity of the PSY enzyme [112]. In addition, phytochromes belong to a family of four members that are ubiquitously expressed in fruits. However, *SlPHYB2* and *SlPHYF* are predominantly expressed in fruits [112,113,114,115,116], whereas *SlPHYA* is mainly expressed in roots, and *SLPHYB1* is mainly expressed in leaves [112,113,114,115,116]. Unexpectedly, only *SlPHYA* increases its transcripts accumulation throughout ripening process, which is particularly dramatic in the columella tissue [117,118]. In fruits, PHYs regulate the cell size, in addition to the expansion and plastids development where chlorophyll and carotenoid synthesis take place [117,118,119,120,121,122]. In fact, *phyaphb1phyb2* triple mutant shows a dramatic reduction in the chlorophyll content in the tomato fruit [122]. Additionally, fruit-localized silencing of *SlPHYA* shows a decrease in chloroplast number and chlorophyll content. Moreover, both *SlPHYA* and *SlPHYB* RNAi plants revealed a decrease in the carotenoid content and expression of *SlPSY1* together with an increase in the relative expression of genes encoding light-repressors, such as *SlCOP1*, DNA damage-binding protein 1 (*SlDDB1*), and detiolated1 (*SlDET1*) [120]. This also correlates with det1 and ddb1 mutants that exert an increase in the size and amount of plastids and chlorophyll and carotenoid content in the fruit [123,124,125,126]. On the contrary, tomato lines carrying the antisense sequence of *LeHY5* produced 24–31% less chlorophyll in leaves compared with control plants and an even greater reduction in chlorophyll and carotenoid accumulation in fruits [123]. On the other hand, fruit-localized overexpression of *SlPHYB2* boosts the carotenoid content and the expression of genes encoding enzymes of the methylerythritol phosphate (MEP) pathway, which provides essential precursors for the carotenoid synthesis, such as 1-deoxy-D-xylulose-5-phosphate synthase 1 (*SlDXS1*) and geranylgeranyl diphosphate synthase 2 (*SlGGDP2*) [121]. The SlPRE2, an atypical bHLH transcription factor, is another negative regulator of the carotenoid biosynthesis that is repressed in high-intensity light in tomatoes. The overexpression of *SlPRE2* promotes hypocotyl elongation and downregulates chlorophyll biosynthesis genes and carotenogenic genes, such as *SlPSY1*, *SlPDS*, and *SlZDS*. Interestingly, expressions of *SlHY5* was also reduced, which could explain the reduction in *SlPSY1* transcripts abundance [127]. Thus, similar to photosynthetic tissue, positive and negative regulators of light signaling are fulfilling similar functions in regulating the synthesis of carotenoids in fruits.

In the mature green (MG) stage, the fruit reaches its full size of development and only contains chloroplast. In the MG stage, the chlorophylls from these chloroplasts absorb blue and R light (450 nm to 650 nm approximately) from the environment, and FR light penetrates the fruit, producing "self-shading" (low R/FR ratio) inside [128]. Therefore, PIF1a binds to the *SlPSY1* promoter, repressing its expression in the flesh of MG (Figure 3). Then, in the breaker (B) stage, when chlorophylls degradation and chromoplast development begins, the proportion of light that reaches the flesh of the fruit is gradually enriched in R (high R/FR ratio), starting the degradation of SlPIF1a. At the same time that the expression of *SlPSY1* increases, the ripening also develops. In the pericarp, the R/FR light ratio is higher, and therefore, the relative level of *SlPSY1* expression and carotenoid synthesis is higher than in the inner section. Once the fruit reaches maturity in the red ripe (RR) stage, the whole fruit (in the flesh and the pericarp) is exposed to a high R/FR ratio, resulting in an enhanced carotenoid synthesis and high accumulation (Figure 3) [1,117,128].

Another less explored model of carotenoid synthesis occurs in non-climacteric citrus fruits. These fruits belong to different species of the *Citrus* genus, which shows a wide range of colorations in its peel and pulp due to the different proportions and composition of carotenoids [129,130,131,132]. In these fruits, light quality affects dramatically the degreening and carotenoid accumulation in the fruit pericarp. Grapefruits (*Citrus paradisi M.*) that are grown under a dense canopy (low R/FR) show a faster degreening process in their peel, resulting in a more intense orange coloration in the peel compared to when they are grown under direct sunlight (high R/FR) (Figure 2) [133]. On the other hand, when satsuma mandarin (*Citrus unshiu M*.) fruits ripe are exposed to R light, they present an increase in the degreening rate, carotenoid content, and in the carotenogenic gene expression, such as *CitPSY*, *CitCRTISO*, *CitLCYb2*, and *CitLCYe* (Figure 3) [134].

Citrus fruits have different types of plastids when comparing pericarp and flesh at the same stage of maturity. In the green (G) or breaker (B) stages of navel orange (*Citrus sinensis O.*), Star Ruby grapefruit (*Citrus paradisi M.*), and Miyagawa wase (*Citrus unshiu M.*), the peel contains chloroplast, accumulating β-carotene and lutein, while the flesh contains chromoplasts that accumulate high levels of carotenoid crystals (Figure 3) [135,136]. It is possible that the chlorophyll in chloroplasts found in the peel absorbs R light, allowing FR light (low R/FR) to penetrate the flesh, generating self-shading and triggering the development of chromoplasts. This phenomenon is also observed in mango and watermelon where chloroplasts are present in the peel of fruits, while chromoplasts develop in the flesh [137,138,139,140].

Citrus fruits also synthesize carotenoids in the absence of direct light. When fruits of Clementine mandarins (*Citrus clementina T.*) and Navelina oranges (*Citrus sinensis O.*) are grown attached to the tree but within black bags that block the light (light avoidance), the degradation of chlorophyll and the synthesis of carotenoid are faster in the pericarp (but not in the flesh) than fruits that are grown under direct W light. However, covered fruits accumulate fewer carotenoids in the peel and have a lower expression of genes involved in the carotenoid biosynthesis, such as *PSY*, in comparison with fruits grown under direct W light [141]. Although the pericarp of covered grapefruits (*Citrus paradisi M.*) produced an increase in carotenoids and chromoplasts differentiation, this does not correlate with an induction in the expression of genes participating in the carotenoid pathway. However, the expression of *HSP20-4* and *HSP21* that codify for chaperons increases their expression, which can be a key factor for chromoplast differentiation [133]. On the contrary, in covered pepper (*Capsicum annuum* L.) and tomato fruits, the content of carotenoids and the expression of *PSY* decreased compared to the fruits grown under direct W light [142]. This confirms that, in these types of fruits, R light has the main role in carotenoid synthesis induction.

### 3.3. Roots: From Leucoplasts to Chromoplasts

Roots are required for anchorage and water and nutrients uptake. Its architecture varies among different plant species and can be modulated by environmental conditions and hormones such as auxin [143]. Usually, the root develops leucoplast or amyloplasts that accumulated negligible levels of carotenoid [144]. In Arabidopsis (*Arabidopsis thaliana* L.) roots, an increase of *AtPSY* expression and carotenoid accumulation is observed after salt exposure. However, the root remains pale because the metabolic flux increases towards the ABA production in order to deal with the abiotic stress [145]. Interestingly, the development of a few chloroplasts has been described in Arabidopsis roots, when they grow under exposure to direct W light for 7–8 weeks after germination. Thus, W light can induce the differentiation of leucoplasts into chloroplasts [146]. This potential of root greening is higher in *Atcop1* and *Atdet1* because of the reduced level of negative regulators of chlorophyll and carotenoid synthesis, showing chloroplasts-like structures that resemble chloroplasts, similar to chloroplasts found in leaves. This evidence reveals that *AtCOP1* and *AtDET1* inhibit chloroplast differentiation also in the roots exposed to W light [146,147] similar to what occurs during photomorphogenesis.

Interestingly, PHYs that act upstream of COP1 and DET1 are expressed in the root grown underground [148], where they participate mainly in the regulation of both lateral and primary root growth [149,150]. Nonetheless, PHYs and CRYs also regulate synergistically the differentiation from leucoplasts to chloroplasts in R and B light, although B light is more effective than R light at inducing the greening in the root. Moreover, in B light, the expression of photosynthetic genes, such as light-harvesting chlorophyll a/b binding protein 1*3 (*Lhcb1*3*) and ribulose-1,5-bisphosphate carboxylase/oxygenase (*RbcS*) is increased, showing that the photosynthetic machinery is actively required for chloroplasts development in the root [151]. Thus, similar to photosynthetic tissue, PHYs and CRYs can positively regulate the development of plastids in roots. Additionally, root greening is also regulated by auxin and cytokinin signaling, which act downstream of PHYs and transcriptional factors, such as HY5 and golden2-like (GLK2). Auxin is synthesized and transported from the shoot to the root of the plant, where it represses root greening, whereas cytokinin is needed for both chlorophyll synthesis and chloroplast development [152,153,154].

Despite the fact that chloroplast development in the Arabidopsis root can be induced under direct W, R, or B light, this is normally not the case, because roots grow underground. Interestingly, in Arabidopsis, it was shown that light sensed in leaves is conducted through the stems and some proportion reaches the roots [148]. Although plants perceive W light, the FR light is more efficiently transmitted than R light through the stems, concluding that roots grow in a context of a very low ratio of R/FR light, which we can define and understand as shade (Figure 3) [148,155,156]. Indeed, under this low R/FR ratio, PHYB is active in roots and regulates growth and gravitropism [148]. However, neither chlorophyll nor carotenoids are normally produced in roots grown underground. Roots that develop chromoplasts and synthesized carotenoids, such as carrot taproot, are scarce [23,145,157,158]. Other common roots that also present a secondary growth together with an accumulation of pigments are beet (*Beta vulgaris* L.) (betalains) and radish (*Raphanus sativus* L.) (anthocyanins) [159,160], but in this study, we focused only on organs that accumulate carotenoids.

In carrots (*Daucus carota* L.), the direct W light affects carotenoid and chlorophyll synthesis in their taproot. W light induces the development of chloroplasts instead of chromoplasts, accompanied by a reduction in carotenoid content and expression of carotenogenic genes, such as *DcPSY1* and *DcPSY2*, compared to the root that grows underground (Figure 3) [2,157]. To elucidate the molecular mechanisms underlying the accumulation of carotenoids in carrot root, an RNA-seq analysis between dark (underground)- and W light-grown carrot roots was carried out. Interestingly, it was determined that light-regulated genes that participate in plastid development and carotenoid synthesis in photosynthetic tissue and fruits, such as *DcPHYA*, *DcPHYB*, *DcPIF3*, *DcPAR1*, *DcCRY2*, *DcFHY3*, *DcFAR1*, and *DcCOP1* were preferably expressed in dark-grown roots and could therefore be involved in the regulation of carotenoid synthesis in the carrot taproot [161]. Therefore, similar to citrus, mango, and melon, the carotenogenic gene expression and carotenoids synthesized in this taproot could be ascribed to the presence of FR light even when it is grown underground.

## 4. Conclusions and Perspectives

Light is a powerful stimulus among the regulatory environmental factors of carotenoid synthesis, which has an increased value due to several studies in the last years, particularly on elucidating the mechanism by which it activates or represses plastid differentiation and carotenoid synthesis.

The knowledge about carotenoid synthesis regulation in plants, including photosynthetic tissues, fruits, and roots, confirms that PSY is the key enzyme in the synthesis of carotenoids, and therefore, the stimuli that regulate carotenoid synthesis ultimately converge on the expression of this key gene.

The perception of R and FR light and its proportions (high or low R/FR) in fruits converge in different carotenoid accumulation and plastids development in citrus and tomato (Figure 3). Plastids are essential to ensure carotenoids accumulation and they determine both the profile and amount of carotenoids in the respective tissue [23].

It is interesting to note that FR light is efficiently transmitted through the stem reaching root cells in Arabidopsis. In this context, the carrot taproot could be in low R/FR ratio condition and not in darkness. This light quality could explain that genes such as *DcPHYA*, *DcPHYB*, *DcPIF3*, *DcPAR1*, *DcCRY2*, *DcFHY3*, *DcFAR1*, and *DcCOP1* are overexpressed in comparison with roots exposed to W light [161]. However, the role and the specific contribution of these genes in the synthesis of carotenoids in the carrot taproot grown underground has to be determined. 

In the same sense, the carrot taproot could resemble the condition of citrus, where FR light that reaches the flesh of the fruit promotes the synthesis of carotenoids together with the expression of *PSY*.

On the other hand, the molecular mechanism that is described in Arabidopsis roots exposed to W light may resemble those shown in carrot taproot when it is grown exposed to W light [2,161]. PHYs and CRYs may induce the differentiation of leucoplast into chloroplasts, inducing taproot greening. In addition, in W light, the expression of photosynthetic genes are upregulated in comparison with dark-grown taproot [161], showing that the photosynthetic machinery is actively required for chloroplasts development in the carrot root [161]. Genes such as *Lhcb*1*3 and *RbcS* expressed in the root of Arabidopsis when it is exposed to light remain to be identified in the carrot transcriptome and further characterization will shed light on this mechanism.

The role of light-activated factors such as *PAR1*, *PHYA*, *PHYB*, and *PIFs* in the development of chromoplasts and accumulation of carotenoids in citrus flesh, tomatoes, and carrots during development and ripening remain to be determined. Considering all the knowledge in shade avoidance in plants, it can be proposed that similar molecular mechanisms are occurring in citrus fruits and carrot taproots, where the FR light is predominant. Thus, PHYA can recruit PIFs together with PAR1, allowing, partially, the binding of HY5, LAF, and other positive transcription factors to genes for carotenoid (*PSY*) and chlorophyll synthesis (Figure 2).

## Figures and Tables

**Figure 1 ijms-22-01184-f001:**
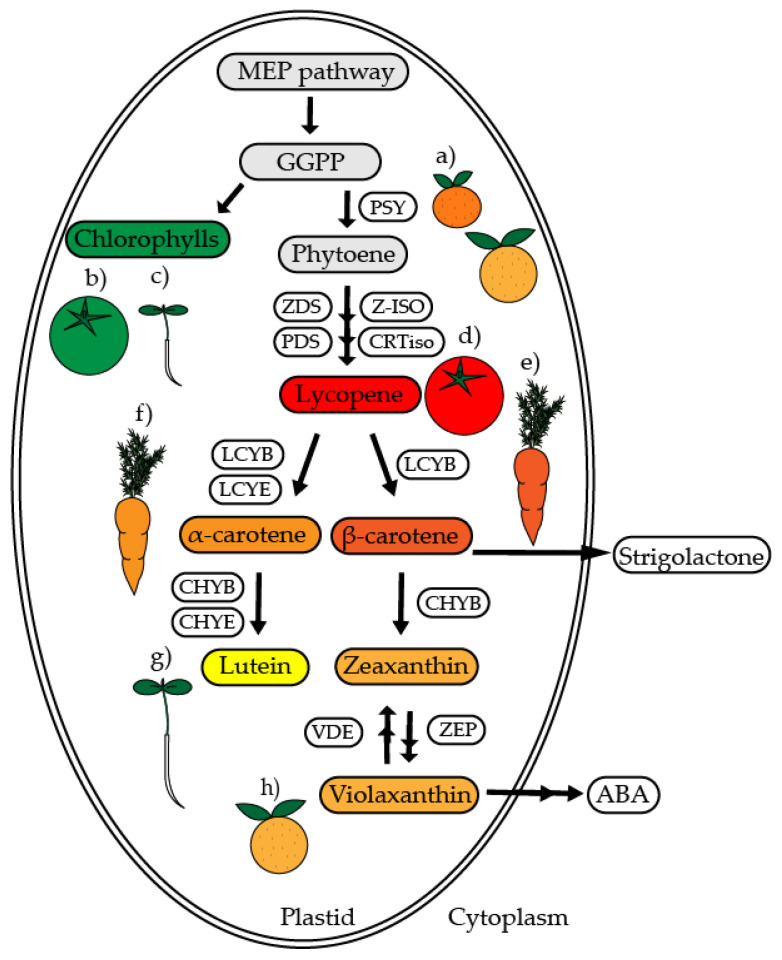
Simplified carotenoid synthesis pathway. Methylerythritol phosphate (MEP) pathway leads to the production of geranylgeranyl pyrophosphate (GGPP), the main precursor of carotenoids and chlorophylls synthesis. We highlighted the main enzymes of the pathway and the most accumulated carotenoids in plant models discussed throughout the review, in (**a**) Star Ruby grapefruits (*Citrus paradisi M*.), Satsuma mandarin (*Citrus unshiu M.*) (**b**) tomato (*Solanum lycopersicum* L.) in mature green (MG) stage, (**c**) leaves, (**d**) tomato in ripening (RR) stage, (**e**,**f**) carrot (*Daucus carota* L.) taproot, (**g**) leaves, and (**h**) Star Ruby grapefruits (*Citrus paradisi M.*). Abbreviations of enzymes: phytoene synthase (PSY), phytoene desaturase (PDS), z-carotene desaturase (ZDS), z-isomerase (Z-ISO), carotenoid isomerase (CRTiso), lycopene β-cyclase (LCYB), lycopene ε-cyclase (LCYE), carotene β-hydroxylase (CHYB), carotene ε-hydroxylase (CHYE), violaxanthin de-epoxidase (VDE), zeaxanthin epoxidase (ZEP).

**Figure 2 ijms-22-01184-f002:**
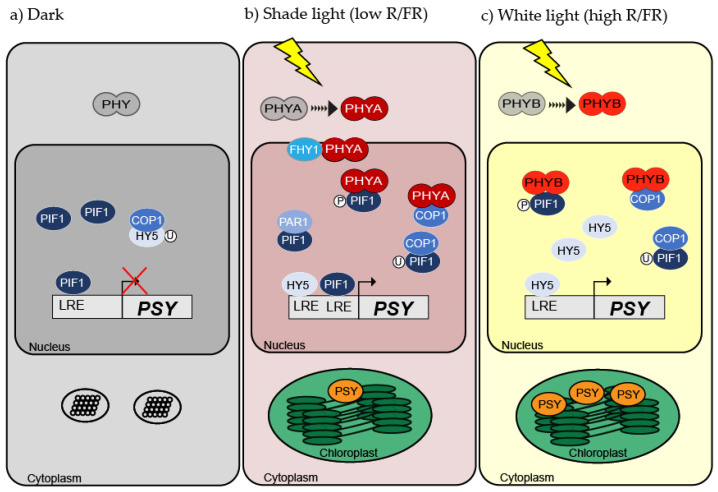
Regulation in carotenoid synthesis in photosynthetic organs under different light conditions. (**a**) Plants grown in dark develop etioplasts were the expression of *PSY* and the synthesis of carotenoids is repressed by phytochrome interacting factors (PIFs) transcription factors. (**b**) Plants grown in shade exhibit a reduced expression of *PSY* and carotenoid synthesis in chloroplasts that take place due to the activation of phytochrome A (PHYA). (**c**) In the presence of white (W) light, a high level of *PSY* expression and carotenoid synthesis is observed due to phytochrome B (PHYB) activation. See further details in the text.

**Figure 3 ijms-22-01184-f003:**
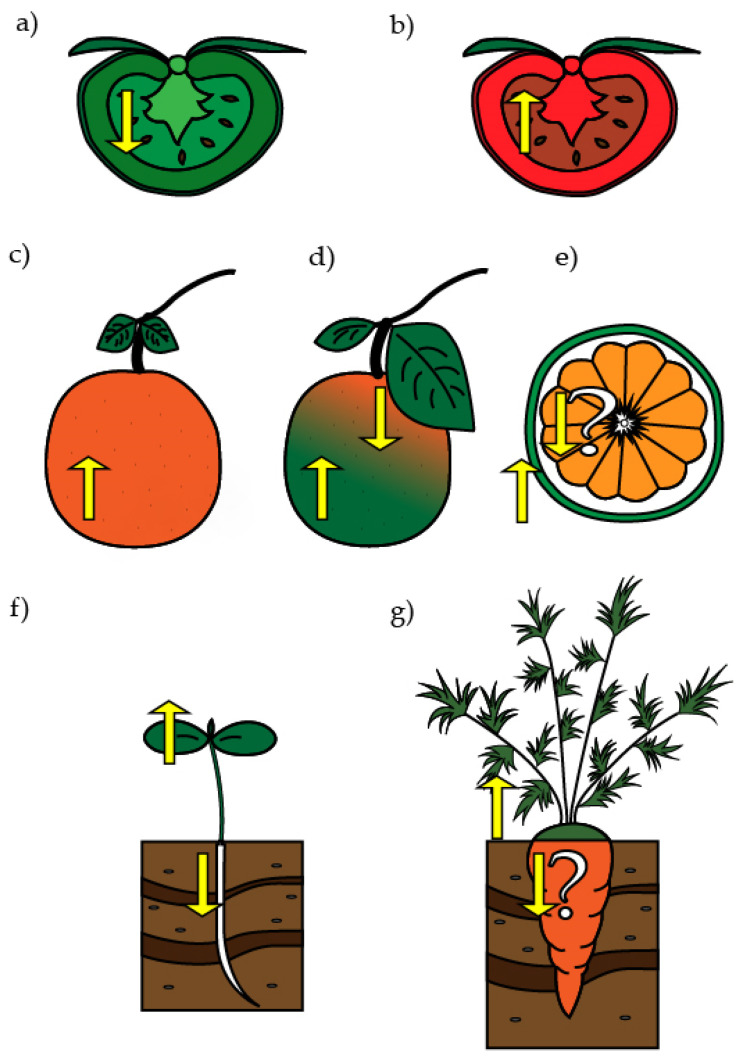
Effect of light on carotenoid synthesis in different plant organs. (**a**) Tomato fruit in MG stage exhibits chloroplasts in low R/FR light. (**b**) Tomato fruit in RR stage presents chromoplasts in high R/FR light. (**c**) Citrus fruit in the ripening stage accumulates carotenoids in high R/FR light. (**d**) Grapefruits accumulate more carotenoids in the pericarp in low R/FR light in comparison with high R/FR light. (**e**) Navel orange (*Citrus sinensis O.*), Star Ruby grapefruit, and Miyagawa wase (*Citrus unshiu M.*) are examples in which, at the breaker (B) stage, carotenoids in chromoplasts are produced in the flesh, whereas in the pericarp, which is exposed to high R/FR, carotenoids are produced in chloroplasts. (**f**) In Arabidopsis (*Arabidopsis thaliana* L.) seedling, high R/FR light induces photomorphogenesis. (**g**) In carrots (*Daucus carota* L.), a high R/FR produces chloroplast development, whilst a low R/FR induces carotenoid accumulation on chromoplasts. Yellow upwards arrows indicate a high R/FR light ratio while downwards arrows indicate a low R/FR light ratio.

## Data Availability

Not applicable.

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
