# Peer review of "Carotenoid Biosynthesis and Plastid Development in Plants: The Role of Light"

_ijms, 2021, doi:10.3390/ijms22031184_

Round 1
Reviewer 1 Report
In the review the effects of light (R, FR, and W) on carotenoid biosynthesis and the role of PHY in photosynthetic tissues, fruits and roots are described. A lot of interesting literature data are not critically discussed but just mentioned as a background for their interpretation taken from the cited papers. I think to understand mechanisms better it is neccessary to include (as taken from the cited papers) quantitative characteristics of incident light intensities: R, FR, W, ratio P/FR at shade, dark, W illumination regimes. In fact in photosynthesis under high light saturation conditions carotenoid synthesis is triggered by plastoquinone reduction. May be PHY operates under much weaker light. Some phrases should be clarified:"incidence of W light in leaves generates the arrival of FR light into root cells", "Chlorophylls trap the R light allowing that FR light reaches the flesh of the fruit". What are the physical mechanisms here? Explane why PHYA and PHYB have different responses to R and FR light.
Author Response
Dear reviewer, see below our response to your comments:
Q1: A lot of interesting literature data are not critically discussed but just mentioned as a background for their interpretation taken from the cited papers. I think to understand mechanisms better it is neccessary to include (as taken from the cited papers) quantitative characteristics of incident light intensities: R, FR, W, ratio P/FR at shade, dark, W illumination regimes.
R1: Dear reviewer, the point you mentioned is very interesting, but it is very difficult to discuss the methodologies of each paper regard light intensities. In general authors used lower intensities of R than FR, but it depends on the specific work and usually authors focused on light qualities more than light intensities. However, in lines 195 we include results of Sng et al (2020), which determined the level of carotenoids in leaves of Arabidopsis plants exposed to white light (R / FR: 3), moderate shade (R / FR: 0.7) and severe shade (R / FR: 0.2). Authors determined that in col-0 plants, carotenoid levels decrease in severe shade, while PSY expression levels decrease in moderate and severe shade. On the other hand, in phya mutants, carotenoid levels decrease in both conditions, although it is more dramatic in deep shade. The conclusion of this work was that PHYA is important to regulate the concentration of carotenoids in the shade, regardless of whether it was moderate or severe, because the focus of the paper is more on the role of PHYA in the synthesis of carotenoids in shade, instead of the dramatic effect of the loss of phya in different shade conditions, so in this case the amount of light does not affect the final result.
Q2: In fact in photosynthesis under high light saturation conditions carotenoid synthesis is triggered by plastoquinone reduction. May be PHY operates under much weaker light.
R2: We totally agree. PHYA can trigger a response in VLFR (Very Low Fluence Response), which is activated by 1-10 µmol m-2 of R light. This was determined in Shinomura, 1996. Briefly, different Arabidopsis plants, either col-0 or mutants for phya and phyb were subjected to R light treatments and the germination percentage was measured. Among the main results, it stands out that at very low amounts of R light (1-10 µmol m-2) germination was inhibited in phya mutants, but not in col-0 or phyb, therefore PHYA can be activated and induced in a very low R light condition. This information was added in line 90 indicating the amounts of light necessary to trigger the response.
Q3: Some phrases should be clarified: "incidence of W light in leaves generates the arrival of FR light into root cells",
R3: In Lee et al., 2016 mentioned this phenomenon as:
“Roots monitor the aboveground light environment by directly sensing stem-piped light under natural conditions”
“Aboveground light is efficiently transmitted through the stems to the roots and that this stem-piped light affects root architecture”
“Light in the FR–near R range was efficiently transmitted through the stems and roots,”
Therefore, we change the phrase for:
“FR light is efficiently transmitted through the stem reaching root cells in Arabidopsis” in line 357.
Q4:"Chlorophylls trap the R light allowing that FR light reaches the flesh of the fruit". What are the physical mechanisms here?
R4: Dear reviewer the physical mechanism derived from this phenomenon is that in the MG stage, chloroplasts accumulate chlorophyll which have the ability to absorb blue light and red light, with peaks close to 450 nm and 650 nm approximately (depending on the type of chlorophyll). Therefore, the wavelengths that are not absorbed by chlorophylls are greater than 700 nm, which generates that in the MG stage the light that is reflected is a large amount of FR light.
Regard your comment, we improve the sentence of line 225 by: In MG stage the chlorophylls from these chloroplasts absorb Blue and R light (450 to 650 nm approximately) from the environment and FR light penetrates the fruit producing "self-shading" (low ratio R/FR) inside (Llorente et al., 2016) and the sentence of line 364 by “In the same sense, the carrot taproot could resembles the condition of citrus, where FR light that reaches the flesh of the fruit promotes the synthesis of carotenoids together with the expression of PSY”.
Q5: Explain why PHYA and PHYB have different responses to R and FR light.
R5: It´s a good point. It is not known for sure why PHYA can respond to R and FR light, while PHYB only to R light. Although PHYA is labile to R light it can certainly be activated with very little R light in VLFR. This is determined by the N-terminal part of the protein, which comprises the N-terminal extension (1-78aa) and the PAS domain (79-185aa) (Oka, et al., 2012). Furthermore, the first 6-12 aa are rich in Serine and could be involved in controlling the degradation of PHYA (Trupkin, et al., 2007). We include a phrase regard this point in line 91.

Reviewer 2 Report
This review is devoted to an interesting and important topic of carotenoid biosynthesis in various plants. The review is well written, structured, and beautifully illustrated. The review will be useful for researchers in this field of science. I believe that the review can be accepted with minor corrections. Minor comments: “Carotenoids are molecules of 40 carbons skeleton” please check this sentence. Carotenoids can be longer or shorter, not all carotenoids are C40 molecules. “they act as accessory pigments in photosynthesis” please clarify the meaning of the word accessory. Scheme of carotenoid biosynthesis and genes involved might be helpful. Please consider that option.Author Response
Dear reviewer, below is the response to your comments:
This review is devoted to an interesting and important topic of carotenoid biosynthesis in various plants. The review is well written, structured, and beautifully illustrated. The review will be useful for researchers in this field of science. I believe that the review can be accepted with minor corrections.
-Thank you very much for your positive comments.
Q1:Minor comments: “Carotenoids are molecules of 40 carbons skeleton” please check this sentence. Carotenoids can be longer or shorter, not all carotenoids are C40 molecules.
R1: We preferably eliminate the idea considering your correct comment.
Q2:“they act as accessory pigments in photosynthesis” please clarify the meaning of the word accessory.
R2: We took a description of “accessory pigments” from Hashimoto et al., 2016 and include some modifications to this sentence in line 38.
“Chlorophylls that play a major role in photosynthesis cannot absorb much light in the 450–550 nm region where the solar radiation profile (spectrum) at the surface of earth has its maximum intensity. This is precisely the region where carotenoids absorb light strongly. They are able to transfer this excitation energy to the chlorophylls, making it available to power photosynthesis. This energy-transfer reaction allows the carotenoids to function as accessory light-harvesting pigments, broadening the spectral range over which light can support photosynthesis (Hashimoto, et al., 2016)”
Q3: Scheme of carotenoid biosynthesis and genes involved might be helpful. Please consider that option.
R3: Yes, of course. We include a carotenoid biosynthesis pathway.
